# Alterations in Intestinal Mucosal Barrier Visualized by Confocal Laser Endomicroscopy in Liver Cirrhosis: A Pilot Trial (AMBIC)

**DOI:** 10.3390/diagnostics14151606

**Published:** 2024-07-25

**Authors:** Monica Alexandrina Rusticeanu, Vincent Zimmer

**Affiliations:** 1Department of Gastroenterology, Hospital Asklepios Klinikum Schwalmstadt, Krankenhausstraße 27, 34613 Schwalmstadt, Germany; 2Department of Gastroenterology, University Hospital Berne, Freiburgstrasse 18, 3010 Bern, Switzerland; 3Department of Medicine, Hospital Knappschaftsklinikum Saar, In d. Humes 35, 66346 Püttlingen, Germany; vincent.zimmer@gmx.de; 4Department of Medicine II, Saarland University Medical Center, Saarland University, 66421 Homburg, Germany

**Keywords:** intestinal permeability, liver cirrhosis, confocal endomicroscopy

## Abstract

Background: Chronic liver disease occurs throughout the world irrespective of region, age, sex, or race, and it is caused by a variety of liver conditions. One of the most frequent infectious complications in liver cirrhosis that severely reduces the median survival is spontaneous bacterial peritonitis. Current guidelines recommend a paracentesis before starting an antibiotic prophylaxis for this complication. Methods: Selective intestinal decontamination significantly lowers the rate of first or recurrent SBP in cirrhotic patients, so in this study we aimed to investigate and quantify the intestinal integrity of patients with liver cirrhosis and correlate a pathologically increased permeability with the incidence of SPB. We included 14 patients who met the inclusion criteria. No patient was excluded. For the CLE investigation, we use probe based confocal laser endomicroscopy techniques from Mauna Kea (Cellvizio), enabling in vivo surface imaging. The images (optical biopsies) were analyzed for functional and structural barrier defects after the procedure using Mauna Kea software (version 1.0.09). Results: Because of the small number of included patients and healthy controls, most results are lacking statistical relevance. We found that the CLE investigation showed an increased intestinal permeability in patients with liver cirrhosis, in concordance with previous published data, based on other assessment methods. Conclusions: This study confirms that previously published permeability scores can be applied for patients with liver cirrhosis and is, to our knowledge, the first to investigate the intestinal permeability in vivo in patients with liver cirrhosis. Further data are needed to identify patients at risk and help develop new and less invasive diagnostic criteria for cirrhotic patients who may profit from a prophylactic antibiotic treatment.

## 1. Introduction

### 1.1. Intestinal Permeability, Bacterial Translocation, and Its Clinical Relevance in Cirrhosis

According to the WHO, about 74% of the mortality is caused by chronic noncommunicable disease [1]. Chronic liver disease occurs throughout the world irrespective of region, age, sex, or race. Cirrhosis is a result of a variety of liver diseases characterized by fibrosis and architectural distortion of the liver leading to the formation of regenerative nodules. This condition can have varied clinical manifestations and complications. Liver disease is responsible for 4% of all deaths, which are most often caused by complications of cirrhosis and hepatocellular carcinoma [2]. The increasing burden of liver disease in the USA led to an estimated 4.5 million diagnosed with liver disease, this being one of the top ten causes of mortality, the second leading cause of mortality amongst all digestive diseases [2].

One widely used modality of assessing severity of liver disease is according to the Child–Pugh–Turcotte (CPT) classification. This score was meant to predict the operative risk in patients with variceal bleeding. It included ascites, hepatic encephalopathy (HE), nutritional status, total bilirubin, and albumin. Pugh et al. added the coagulation parameters and removed nutritional status [3].

Prognosis worsens with increasing Child–Pugh score. In addition, bacterial infections are the main drivers of mortality in cirrhosis.

Indeed, spontaneous bacterial infections are up to 10 times more frequent in cirrhotic patients than in liver-healthy individuals and increase mortality by a factor of 4 [4,5]. A frequent infection in cirrhosis is the so-called spontaneous bacterial peritonitis (SBP) [6]. SBP is associated with significant mortality, estimated in the literature between 15 and 40% [7].

Its pathogenesis involves bacterial translocation, with bacteria entering the lymphatic circulation and the bloodstream; this is eased by an impaired immune system and low defense mechanisms such as an impaired intestinal barrier and increased permeability [8,9,10]. Most importantly, pBT occurs in conditions of decompensated cirrhosis presenting with ascites [10,11].

Moreover, the only study available in humans evaluating pBT in cirrhosis observed an increase in culturable bacteria in mesenteric lymph nodes only in advanced stages of the disease, classified as Child Class C [10]. Risk factors for SBP besides advanced stage of disease are (i) the occurrence of a gastrointestinal bleed hampering splanchnic perfusion and increasing pathological bacterial translocation (pBT) and (ii) a low total protein content in ascites (<1.5 g/dL). The latter reflects decreased opsonic and complement levels [8]. Previous research found that, at total protein concentrations >1.5 g/dL, the 1-year rate for SBP is only 1–2%. In contrast, at levels <1.5 g/dL or even 1.0 g/dL, the corresponding rates are about 15–20% and 20–30%, respectively [12].

SBP is caused mostly by enteric organisms via pathological bacterial translocation (pBT) [11,13,14]. Previous trials demonstrated that selective intestinal decontamination significantly lowers the rate of first or recurrent SBP in cirrhotic patients [15].

This clearly underlines the therapeutic potential of addressing the intestinal mucosal barrier in cirrhosis.

Antibiotic treatment targeting intestinal bacteria is effective in preventing SBP and is indicated in:Patients with cirrhosis and gastrointestinal bleeding [16,17].Patients with cirrhosis and ascites if the ascitic fluid protein is <1.5 g/dL (15 g/L) along with either impaired renal function or liver failure. Impaired renal function is defined as a creatinine ≥1.2 mg/dL (106 micromol/L), a blood urea nitrogen level ≥25 mg/dL (8.9 mmol/L), or a serum sodium ≤130 mEq/L (130 mmol/L]). Liver failure is defined as a Child–Pugh score ≥9 and a bilirubin ≥3 mg/dL (51 micromol/L).Patients who have had one or more episodes of SBP. In such patients, recurrence rates of SBP within one year have been reported to be close to 70 percent if no antibiotic prophylactic measure is initiated [5,18].

### 1.2. Confocal Laser Endomicroscopy (CLE): Description of the Device and Method, Indications, and Prior Studies

The CLE technique is based on tissue illumination after systemic application of fluorescein sodium in a dilution of 10%, which highlights the extracellular matrix [19].

We used the so-called probe-based CLE (pCLE) from Cellvizio 100 Series by Mauna Kea, Mauna Kea Technologies, 9 rue d’Enghien, 75010 Paris, France. This method uses a fibered probe, which is passed through the working channel of a standard gastroscope or colonoscope [20]. This system is intended to allow confocal laser imaging of the internal microstructure of tissues within or adjacent to anatomical tracts, i.e., gastrointestinal (but also urinary or respiratory) accessed through an endoscope.

Confocal miniprobes are made of a distal tip, a fiber with protective sheath, and a connector. When connected to the Cellvisio system, they transport the scanned laser beam through the sheated fiber to the site of observation in contact with the distal tip and capture the fluorescent light emitted back from the tissue to provide in vivo fluorescence imaging of tissues. The parts of the confocal miniprobes that come in contact with the patient consist of biocompatible glass, plastic, and metal derivatives.

The system uses a 488 nm wavelength laser. The confocal images are steamed at a frame rate of 12 frames per second, obtaining real-time videos of the mucosa [21].

The contrast agent, fluorescein, has been used as a nontoxic agent in ophthalmology for a long time (for retinal angiography), and multiple independent studies have highlighted its safety for endomicroscopy [19].

The CLE method permits in vivo microscopy of the human gastrointestinal mucosa during endoscopy, providing optical virtual biopsies. Endomicroscopy has been largely used in a multitude of diseases, with most studies focusing on inflammation and neoplasia, such as Barrett’s esophagus, gastric cancer, celiac disease, inflammatory bowel disease, or colorectal neoplasia. It facilitates the study of pathophysiological events in their natural environment (functional imaging) [22,23,24,25,26].

To our knowledge, the role of endomicroscopy in patients with liver disease was confined to imaging the biliary tree [27] but has not been used so far to evaluate the intestinal mucosa of these patients. At present, sugar absorption tests (lactulose and mannitol) are used to assess intestinal permeability and, hence, mucosal barrier function of the intestinal tract. There are no data at this point that show in vivo signs of intestinal barrier dysfunction in liver disease by means of confocal endomicroscopy. In this study, we aim to investigate and quantify the intestinal integrity of patients with liver cirrhosis, thus improving our understanding of this gastrointestinal pathophysiology at the cellular level.

## 2. Study Objectives

The overall objective of this observational study was to evaluate the confocal laser endomicroscopy (CLE) technique with Cellvizio in the setting of endoscopy and to define parameters that are altered in cirrhotic patients of different severity and being at risk of developing a SBP (spontaneous bacterial peritonitis). These preliminary data would support a future trial design aiming at establishing and validating CLE-based risk factors for spontaneous bacterial peritonitis in cirrhosis. The primary objective, which was meant to define a correlation between altered permeability parameters and the risk of developing spontaneous bacterial peritonitis, could not be met because of insufficient follow-up. Also, of the included patients with ascites (4), none had a medical indication for ascites fluid preservation.

## 3. Investigational Plan

We aimed to compare in this 5-arm prospective pilot study patients with and without liver cirrhosis intended for endoscopy.

We enrolled 15 patients who met the inclusion criteria and to whom none of the exclusion criteria applied.

Eligible subjects were, according to the study protocol, men and women aged 18 and older with liver cirrhosis of any cause and healthy controls undergoing elective endoscopy.

Exclusion criteria were significant physical limitations that could affect the outcome, pregnant or breastfeeding women, known allergy to fluorescein, reported cardiac disease or bronchial asthma, contraindication for an endoscopic examination, participation in another clinical investigation or study, or persons who have been placed in an institution because of an administrative or judicial order.

Measurements were obtained at inclusion with respect to clinical and laboratory parameters. We registered the following biological parameters:

CPT-stadium, ascites, serum Na, serum bilirubin, NAFLD, gender, etiology of the cirrhosis, encephalopathy grade, presence of SBP, varices, presence of portal vein thrombosis, serum albumin, serum creatinine, CRP, quick time, hematocrit, thrombocytes, co-morbidities such as arterial hypertension (AHT) and diabetes, as well as intake of medication such as diuretics, antibiotics, PPI, beta-blockers, or other medication.

We performed CLE of the duodenum during a gastroscopy, after the application of a 5 mg fluorescein bolus. The images were registered immediately after bolus application and were interpreted afterwards. The investigators were blinded regarding the cirrhosis stage during the interpretation of the images.

Unfortunately, only one healthy control could be enrolled to compare the permeability of the healthy intestine and the permeability modifications in the different subgroups. Follow-up at 1 year could not be achieved (Figure 1).

## 4. Materials and Methods

### 4.1. Patients

We were able to enroll 15 patients between September 2017 and December 2019. One patient was a healthy control, one patient had portal vein thrombosis, and the other 13 patients had liver cirrhosis. The Child–Pugh score was between A (5) and C (13).

We added 7 control patients without liver cirrhosis and intact permeability parameters from a retrospective cohort in Asklepios Klinikum Schwalmstadt, Germany, an academic hospital of the University Marbug-Gießen. These patients underwent CLE (Cellvizio 100 Series, Mauna Kea Technologies, 9 rue d’Enghien, 75010 Paris, France) investigation of the duodenum before CLE food allergy testing between 2021 and 2023. The retrospective analysis of the images was approved by the local ethics committee.

Demographic data: 13 patients were male, aged from 23 to 77 years; 9 patients were female, aged from 21 to 73 years.

Etiology of liver cirrhosis: NAFLD 7 patients, viral 3 patients, ASH 3 patients, other/unknown 1 patient.

Encephalopathy: 3 patients had hepatic encephalopathy grade 1 at the time of the CLE investigation.

Ascites: 4 patients had ascites. None of the patients had spontaneous bacterial peritonitis at the time of the investigation or in their medical history.

Varices: 12 patients had esophageal varices; 8 patients were on beta-blocker medication.

The complete red-cap data can be accessed at any time.

### 4.2. CLE Interpretation Method

CLS (confocal leak score): We used the CLS score as described and published by Jeff Chang et al. [28]. The stored images were evaluated by one blinded, experienced researcher for the presence or absence of signs of increased intestinal permeability. Inadequate images were excluded (blurred or with detectable villi in less than 1/3 of the surface). The images were then divided into two categories (positive, with signs of increased permeability, and negative, without such features). The images were interpreted as positive if they had 1 or more of the following: cell junction enhancement, fluorescein leak, or cell drop out. Identical features on serial pictures were counted to a maximum of 5 times. The CLS (confocal leakage score) was calculated by dividing the number of positive images by the total number of images. A minimum of 65 images per patient was analyzed with this method.

Watson grade: Watson grade was defined according to the publication from Lee et al. [29] by evaluating the cell shedding, the micro erosions (with exposed lamina propria to the lumen), and the presence or absence of fluorescein signal in the intestinal lumen. The Watson classification contains 3 grades: I (normal barrier function) with cell shedding confined to single cells per shedding site without local barrier disfunction; II (functional defect) with cell shedding confined to single cells per shedding site, fluorescein leakage into the lumen, III (structural defect) with shedding of multiple cells per shedding site, with exposure of the lamina propria and fluorescein leakage into the lumen.

Gap density: We evaluated the gap density according to the publication by Liu JJ et al. [30] by counting the cells and gaps in adequately imaged villi and calculating the density per 1000 epithelial cells. Villi images were considered adequate when at least 75% of the surface was visualized in the pCLE images on 3 consecutive images.

Statistical analysis was performed by NOVUSTAT. Because of the limited number of enrolled patients, the preferred statistical method was descriptive.

The statistic tables contain the number of valid observations (*n*), the mean (M), the standard deviation (SD), as well as the minimum (min) and maximum (max) for scale variables. For categorical variables, the absolute and relative frequencies are given for each category. For comparing scale variables, we performed the two-sample *t*-test for independent samples; for comparing categories, we used either the chi square test of independence or Fischer’s exact test, depending on the number of possible categories. The Pearson correlation was used as the correlation coefficient. The software used for the evaluation was R (version 4.0.2).

## 5. Results

Because of the small size of the cohort, most results are not statistically significant.

11.1. Intestinal permeability measured by the CLS score: The only statistically significant correlation was between the degree of liver damage (cirrhosis) and the CLS score. We found that the CLS score was significantly higher in patients with liver cirrhosis than in patients without liver cirrhosis (4.5 for CTP A, 3.4 for CTP B + C vs. 0.5 for “no” CTP, *p* = 0.001). This finding confirms previously published data about the increased intestinal permeability in patients with advanced liver disease. There was no significant difference in the CLS score between patients with compensated cirrhosis (CTP A) and decompensated disease (CTP B + C).

11.2. Watson grade: Patients with a highly disturbed permeability (Watson Grade III) had a significantly higher CLS score when compared to the normal or intermediate grade (Watson grades I and II) (mean 3.7 vs. 1.5, *p*: 0.042).

Other findings without statistical significance:

11.3. Gap density: The gap density rises in our patients’ evaluation in concordance with the CTP stadium, being lowest for child A and highest for child C patients with an average of 119.50 for child A, 127.09 for child B, and 134.03 for child C cirrhosis. (*p*: 0.960)

If we were to find a cutoff for the presence of liver cirrhosis as predicted by the intestinal permeability (defined as gap density) measured by Cellvizio, this would be 95.6. This means that for our investigation, all patients with a gap density under 95.6 do not have liver cirrhosis; all patients with liver cirrhosis have a gap density over 95.6.

The three CLE-permeability scores used in our interpretation of the data were convergent; for Watson scores I and II, we measured a lower gap density and CLS score; for Watson score III, higher values for gap density and CLS (103.5/1.5 and 117.5/3.7, respectively).

## 6. Discussion and Overall Conclusions

Our small pilot study shows that patients with liver cirrhosis have increased intestinal permeability as defined by standardized and previously published CLE—scoring systems. All evaluation modalities (Watson grade, gap density, and CLS) showed increased intestinal permeability for cirrhosis patients (as defined by previous publications compared to the general population/controls without liver cirrhosis), consistent with prior research results.

Although without statistical relevance, we found that the degree of liver damage (and an advanced Child–Pugh score) was in concordance with the degree of permeability, with the latter rising in patients with decompensated disease and a higher score.

Our investigation provided a good correlation between the currently used CLE scores evaluating the intestinal permeability and showed that CLE may become a feasible method for assessing intestinal permeability in patients with liver disease, an important puzzle piece in the complex mechanism that leads to decompensation of the liver function.

The interpretation of the intestinal permeability as provided by endomicroscopic imaging can be challenging since the intestinal barrier is not uniform and, as we found, scores of intermediate impairment (like Watson II) are present in healthy controls. The accuracy of the assessment is rising with the number of images analyzed, so that could be an explanation to why the CLS score was performing best in differentiating cirrhotic patients with disturbed permeability from healthy controls. Also, observation of the image dynamics is important (video capture), since the system is adapting the contrast automatically, and differences in luminosity can be observed in adjacent images, so determination of leakage can be challenging, particularly for the gap density score.

Emerging data are underlining the importance of intestinal permeability in different conditions, and targeting intestinal permeability directly might be a treatment approach in the future. More data describing variations in intestinal permeability assessed with CLE and pathophysiological correlations will help understand how the gut and its content are regulating health.

Patients with liver cirrhosis are frequently subjects of endoscopic investigations, since the portal hypertension may lead to esophageal varices, and this condition needs follow-up and prophylactic therapy. Assessing the gut barrier integrity with CLE could be integrated into this evaluation, providing additional information and completing the risk constellation of these patients.

Data support the theory that rifaximine, which is already used for the treatment of SBP, might modulate intestinal barrier permeability with only a minor influence on the intestinal microbiota [31]. Other substances like lecithin [32] and the composition of the intestinal mucus play an important role in the integrity of the intestinal barrier. Future efforts are warranted to address the intestinal barrier function and specific therapies in patients with liver cirrhosis. More research is needed concerning the predictive value of the permeability evaluation (for the development of SBP) to provide the best prophylactic strategy.

## Figures and Tables

**Figure 1 diagnostics-14-01606-f001:**
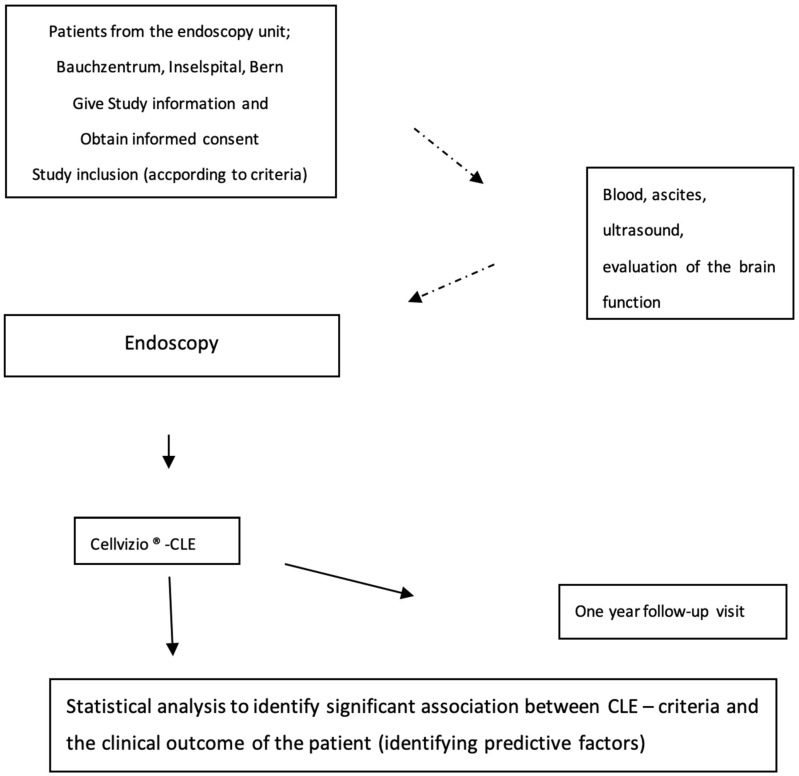
Flow chart of the study: the 5 subgroups could not be formed as initially planned due to the small number of enrolled patients.

## Data Availability

Study data is available in RED Cap-CTU—Inselspital Berne.

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
