# Peer review of "Alterations in Intestinal Mucosal Barrier Visualized by Confocal Laser Endomicroscopy in Liver Cirrhosis: A Pilot Trial (AMBIC)"

_diagnostics, 2024, doi:10.3390/diagnostics14151606_

Round 1

Reviewer 1 Report

Comments and Suggestions for Authors

pBT is a term not defined in text. I suggest a flow chart for study design. The section of discussion and results can be improved. As the authors say, the statistical impact of the results is low. I suggest an analysis of a statistically significant batch. More confocal microscopy images and their comments are needed.

Comments on the Quality of English Language

Author Response

pBT is a term not defined in text. I suggest a flow chart for study design. The section of discussion and results can be improved. As the authors say, the statistical impact of the results is low. I suggest an analysis of a statistically significant batch. More confocal microscopy images and their comments are needed."

Answers to the reviewer:

pBT is a term not defined in text. Thank you very much for the comment, pBT means pathological bacterial translocation. That was changed in the manuscript. I suggest a flow chart for study design. We agree and a flowchart was added as a separate figure.The section of discussion and results can be improved. Thank you, we revised the discussion. As the authors say, the statistical impact of the results is low.I suggest an analysis of a statistically significant batch. Thank you for your comment, the study is meant to be a pilot study, “proof of concept”, of course more research is needed in functional imaging. Since the method is new, data acquisition can be challenging, nevertheless we plan to conduct further research on this subject.

More confocal microscopy images and their comments are needed. We added more images and comments.

Reviewer 2 Report

Comments and Suggestions for Authors

Dr. Rusticeanu  and Zimmer,

Thank you. I had the opportunity in reading your pilot study related to Alterations in Intestinal Mucosal Barrier Visualized by the application of probe-based Confocal Laser Endomicroscopy by Cellvizio in Cirrhotic patients. 

I have a question in your selection pf patients;

1. Did you have any patients with significant alcohol history?

2. It appears that 50% of the study subjects were metabolic liver disease.

3.Did beta blocker therapy make any difference in permeability?

Best wishes

Author Response

Dr. Rusticeanu  and Zimmer,

Thank you. I had the opportunity in reading your pilot study related to Alterations in Intestinal Mucosal Barrier Visualized by the application of probe-based Confocal Laser Endomicroscopy by Cellvizio in Cirrhotic patients. 

I have a question in your selection pf patients;

1. Did you have any patients with significant alcohol history?

2. It appears that 50% of the study subjects were metabolic liver disease.

3.Did beta blocker therapy make any difference in permeability?

Answers for the reviewer

  1. Did you have any patients with significant alcohol history? Thank you for your question, 3 of the patients had an alcohol induced liver cirrhosis
  2. It appears that 50% of the study subjects were metabolic liver disease. Thank you, that is correct.

3. Did beta blocker therapy make any difference in permeability? This is a very good and interesting point, there was no statistical evidence with regards to the use of betablockers. This may change once more data is gathered in the future. 

Reviewer 3 Report

Comments and Suggestions for Authors

Diagnostics. Title: Alterations in Intestinal Mucosal Barrier Visualized by 2 Confocal Laser Endomicroscopy in Liver Cirrhosis: A Pilot 3 Trial (AMBIC). The concept is good but the included patient population is too small to make any meaningful interpretation. The authors should include more patients and controls and publish the data. My comments are as follows:

1.      The study is a pilot study.

2.      Use full forms at first use of abbreviations (pBT)

3.      Clarify the term CPT-Stadium

4.      The authors should include more patients and controls and publish the data.

5.      Were there any differences based on the etiology of cirrhosis

6.      Hepatic encephalopathy grades of the patients?

7.      The small numbers prevent any kind of statistical analysis. The authors should remove all comparisons- between cirrhosis and no-cirrhosis - cut-off proposed from the analysis as comparisons between such small samples are associated with errors.

8.      The small sample size is a major limitation of the study. Can the authors explain why such a low number of patients were recruited?

Comments on the Quality of English Language

None

Author Response

Diagnostics. Title: Alterations in Intestinal Mucosal Barrier Visualized by 2 Confocal Laser Endomicroscopy in Liver Cirrhosis: A Pilot 3 Trial (AMBIC). The concept is good but the included patient population is too small to make any meaningful interpretation. The authors should include more patients and controls and publish the data. My comments are as follows:

  1. The study is a pilot study.
  2. Use full forms at first use of abbreviations (pBT)
  3. Clarify the term CPT-Stadium
  4. The authors should include more patients and controls and publish the data.
  5. Were there any differences based on the etiology of cirrhosis
  6. Hepatic encephalopathy grades of the patients?
  7. The small numbers prevent any kind of statistical analysis. The authors should remove all comparisons- between cirrhosis and no-cirrhosis - cut-off proposed from the analysis as comparisons between such small samples are associated with errors.
  8. The small sample size is a major limitation of the study. Can the authors explain why such a low number of patients were recruited?

Answers to the reviewer:

  1. The study is a pilot study. Thank you, we agree, this is stated in the discussion. 
  2. Use full forms at first use of abbreviations (pBT) - Thank you for the correction, the sentence was corrected.
  3. Clarify the term CPT-Stadium – Thank you, this is a very good point, following text has been added: Child-Pugh-Turcotte (CPT) classification. This score was ment to predict the operative risk in patients with variceal bleeding. It included ascites, hepatic encephalopathy (HE), nutritional status, total bilirubin, and albumin. Pugh et al added the coagulation parameters and removed nutritional status.

  1. The authors should include more patients and controls and publish the data. Thank, you, indeed, more research is needed in functional imaging, the study is meant to be a pilot study, “proof of concept”. Since the method is new, data acquisition can be challenging, nevertheless we plan to conduct further research on this subject.

  1. Were there any differences based on the etiology of cirrhosis No, our statistical evaluation did not show any differences in permeability, based on etiology.
  2. Hepatic encephalopathy grades of the patients? Thank you, all patients with HE had grade 1 encephalopathy, this information was added in the manuscript.
  3. The small numbers prevent any kind of statistical analysis. The authors should remove all comparisons- between cirrhosis and no-cirrhosis - cut-off proposed from the analysis as comparisons between such small samples are associated with errors. Thank you very much for your comments, we added more non-cirrhotic patients with intact permeability to emphasize the changes found in liver cirrhosis.
  4. The small sample size is a major limitation of the study. Can the authors explain why such a low number of patients were recruited? Thank you, the patients were recruited over a period of two years, for as long as the personal resources conducting the investigation could be sustained. The number of patients resulted after obtaining informed consent and applying the inclusion / exclusion criteria to the cirrhotic patients in the study center.

Reviewer 4 Report

Comments and Suggestions for Authors

The syndrome of leaky gut is well known pathology in cirrhotic patients. Nevertheless, we are still unable to use it in everyday clinical practice. Thus, attempts made to fill this gap are very important. The authors of this manuscript decided to use endomicroscopy to assess the integrity of the wall of the gastrointestinal tract. A small number of patients were included in the study, nonetheless the idea is quite pioneering. In the abstract (lines 19-20) authors wrote about the correlation between the development of SBP among examined patients and disrupted intestinal barrier. Simultaneously, in the results it is described that permeability was just increased in investigated cirrhotic patients. Therefore, it should be modified. Furthermore, I would like authors to mention some kind of serological markers of intestinal integrity; were they explored in cirrhotic patients? Additionally, the expression in line 63 (ii) should be explained.

Author Response

The syndrome of leaky gut is well known pathology in cirrhotic patients. Nevertheless, we are still unable to use it in everyday clinical practice. Thus, attempts made to fill this gap are very important. The authors of this manuscript decided to use endomicroscopy to assess the integrity of the wall of the gastrointestinal tract. A small number of patients were included in the study, nonetheless the idea is quite pioneering. In the abstract (lines 19-20) authors wrote about the correlation between the development of SBP among examined patients and disrupted intestinal barrier. Simultaneously, in the results it is described that permeability was just increased in investigated cirrhotic patients. Therefore, it should be modified. Furthermore, I would like authors to mention some kind of serological markers of intestinal integrity; were they explored in cirrhotic patients? Additionally, the expression in line 63 (ii) should be explained.

Answers to the reviewer:

The syndrome of leaky gut is well known pathology in cirrhotic patients. Nevertheless, we are still unable to use it in everyday clinical practice. Thus, attempts made to fill this gap are very important. The authors of this manuscript decided to use endomicroscopy to assess the integrity of the wall of the gastrointestinal tract. A small number of patients were included in the study, nonetheless the idea is quite pioneering. In the abstract (lines 19-20) authors wrote about the correlation between the development of SBP among examined patients and disrupted intestinal barrier.

Thank you very much for your review and comments. We mentioned this correlation because it was the main idea of the study, unfortunately we could not follow up the patients for this end point, nevertheless, we think that this should be the subject of further research, because it would have therapeutic implications, and this is the reason we mentioned it.

Simultaneously, in the results it is described that permeability was just increased in investigated cirrhotic patients. Therefore, it should be modified. Furthermore, I would like authors to mention some kind of serological markers of intestinal integrity, were they explored in cirrhotic patients?

We considered comparing our CLE results with classical permeability markers, like blood zonulin and lactose mannitol test, but unfortunately, we were not able to implement these investigations for the study patients.

Additionally, the expression in line 63 (ii) should be explained.

pBT pathological bacterial translocation, the sentence was completed accordingly

Round 2

Reviewer 1 Report

Comments and Suggestions for Authors

the requested changes were made according to the mentioned indications. In my opinion, no other changes are necessary. 

Reviewer 3 Report

Comments and Suggestions for Authors

None